# Optimal Tailoring of CNT Distribution in Functionally Graded Porous CNTRC Beams

**DOI:** 10.3390/polym15020349

**Published:** 2023-01-09

**Authors:** J. R. Cho, H. J. Kim

**Affiliations:** 1Department of Naval Architecture and Ocean Engineering, Hongik University, Jochiwon, Sejong 30016, Republic of Korea; 2Department of Mechanical Engineering, University College London, London WC1E7JE, UK

**Keywords:** porous CNTRC beams, functionally graded, thickness-wise CNT distribution, multi-objective optimization, deflection and effective stress, exterior penalty-function method

## Abstract

This paper is concerned with the multi-objective optimization of thickness-wise CNT distribution in functionally graded porous CNT-reinforced composite (FG-porous CNTRC) beams. The mechanical behaviors of FG-porous CNTRC structures are strongly influenced by the thickness-wise distributions of CNTs and porosity. Nevertheless, several linear functions were simply adopted to represent the thickness-wise CNT distribution without considering the porosity distribution, so these assumed linear primitive CNT distribution patterns are not sufficient to respond to arbitrary loading and boundary conditions. In this context, this study presents the multi-objective optimization of thickness-wise CNT distribution in FG-CNTRC porous beams to simultaneously minimize the peak effective stress and the peak deflection. The multi-objective function is defined by the larger value between two normalized quantities and the design variable vector is composed of the layer-wise CNT volume fractions. The constrained multi-objective optimization problem is formulated by making use of the exterior penalty-function method and the aspiration-level adjustment. The proposed optimization method is demonstrated through the numerical experiments, and the optimization solutions are investigated with respect to the porosity distribution and the combination of aspiration levels for two single-objective functions. It is found from the numerical results that the optimum CNT distribution is significantly affected by the porosity distribution. Furthermore, the proposed method can be successfully used to seek an optimum CNT distribution within FG-porous CNTRC structures which simultaneously enhances the multi-objective functions.

## 1. Introduction

Functionally graded carbon nanotube-reinforced composites have been spotlighted as a state-of-the-art composite due to their excellent mechanical properties [1,2]. These advanced composites were developed based on the excellence of carbon nanotubes (CNTs) and the notion of functionally graded materials (FGMs) [3]. The mechanical strength of polymer composites increases dramatically when only a small amount of CNTs are inserted [4], so that CNT-reinforced composites (CNTRCs) can provide higher bending stiffness and superior free vibration and buckling behaviors than the conventional polymer composites [5,6,7,8]. In mechanical applications, CNTRCs have been produced in the form of beams, plates, and shells which exhibit the thickness-wise variation in their mechanical behaviors. Hence, the uniform distribution of CNTs through the thickness may not be appropriate to respond to such thickness-wise variation. According to Seidel and Lagoudas [9] and Qian et al. [10], it has been reported that the enhancement of mechanical properties of CNTRCs is limited when CNTs are uniformly distributed through the thickness. To overcome this limitation, Shen [11] and Ke et al. [12] proposed the purposeful thickness-wise CNT distributions by utilizing the notion of FGM, which is characterized by the continuity and functional tailoring of thickness-wise material composition distribution [13].

Since then, functionally graded CNT-reinforced composites (FG-CNTRCs) have been used to indicate the CNTRCs with purposeful thickness-wise CNT distributions. Among the representative ones with such purposeful thickness-wise CNT distributions are FG-U, FG-V, FG-O, and FG-X. Additionally, extensive research efforts have focused on investigating the mechanical behaviors of FG-CNTRCs, particularly to examine the characteristics of such primitive CNT distribution patterns. The reader may refer to a paper by Liew et al. [14] for a broad literature survey on the studies of FG-CNTRC structures. In the early studies of FG-CNTRCs, the majority of research dealt with the FG-CNTRCs that do not include porosity. However, it was revealed that the difference in solidification temperatures of constituent materials during the fabrication may cause porosities [15,16]. So, more recently, a number of attempts to model and investigate porous FG-CNTRCs have been presented. The effective material properties of porous materials were predicted using a simplified mixture rule, and Ji et al. [17] validated the simplified mixture rule such that the predicted effective material properties coincide well with the experiment.

According to our literature survey, Zhou et al. [18] numerically analyzed the effects of the preload on the thermomechanical behavior and service reliability of the porous FG bolted joint. Chen et al. [19] investigated the nonlinear free vibration behavior of a functionally graded porous sandwich beam using Timoshenko beam theory and the Ritz method. Yang et al. [20] investigated the buckling and free vibration behaviors of FG-porous CNT-reinforced nanocomposite plates using the first-order shear deformation theory and the Chebysheve–Ritz method. Dong et al. [21] investigated free vibration characteristics of FG-graphene-reinforced porous nanocomposite shells with spinning motion using the first-order shear deformation theory. Medani et al. [22] investigated the static and dynamic behavior of FG-CNT-reinforced porous sandwich plates using the first order shear deformation theory and Hamilton’s principle. Setoodeh et al. [23] analyzed the free flexural vibration behavior of doubly curved sandwich shells with FG porous core using the general higher-order shear deformation theory and the generalized differential quadrature method. Polit et al. [24] presented the static bending and elastic stability analyses of thick FG-porous CNTRC curved beams using a higher-order shear deformation theory by considering the through-thickness stretching effect. Hamed et al. [25] investigated and optimized critical buckling loads of sandwich FG beams with a porous core by adopting the parabolic higher-order shear deformation theory. Anamagh and Bediz [26] investigated the vibration and buckling behavior of FG-porous CNTRC plates using the first shear deformation theory and spectral Chebyshev approach. Ebrahimi et al. [27] investigated the free vibration response of sandwich plates with porous electro-magneto-elastic FG facesheets and FG-CNTRC core using a four-variable shear deformation refined plate theory. Madenci and Ozkili [28] explored the influence of porosity on free vibration analysis of FG beams with different boundary conditions using different analytical and numerical approaches. Babaei et al. [29] performed the dynamic analysis of FG-saturated porous rotating thick truncated cones using the graded finite elements and Newmark method. Dat et al. [30] numerically investigated the influence of CNTs, porosity, and thermo-mechanical loading on the vibration and dynamic response of the sandwich FG-CNTRC composite plates based on the higher-order shear deformation theory. For a more detailed literature survey, refer to a review paper by Barbaros et al. [31].

In the open literature, most studies were concerned with the investigation of mechanical behaviors of FG-porous CNTRC structures with respect to the major parameters such as the CNT distribution pattern. These parametric studies are surely helpful in selecting an appropriate one from the above-introduced primitive CNT distributions, but nevertheless, this method can be only considered as a passive one because the suitability of CNT distribution pattern is affected by the structure geometry and the loading/boundary conditions. The reason is because such geometry and conditions definitely influence the mechanical behavior of functionally graded structures [32]. Therefore, the development of an active method for freely tailoring a best suitable CNT distribution pattern is a prerequisite for the success of desired application of FG-CNTRCs. In this regard, this study aims at developing a multi-objective optimization method for seeking a best suitable thickness-wise CNT distribution for FG-porous CNTRC structures.

As an extension of our previous work [33], the optimization problem is defined by the minimization of the peak deflection and the peak effective stress of FG-CNTRC beams at the same time by considering the porosity. The relatively larger values between these two peak quantities are defined by the multi-objective function, and the thickness-wise porosity distribution is modeled by adopting cosine functions. An FG-porous CNTRC beam is uniformly divided into a finite number of sub-layers with layer-wise uniform CNT volume fractions, in order to suppress the increase of design variable number and maintain the CNT distribution flexibility at the same time. The CNT volume fractions of sub-layers are defined as the design variables and subjected to the inequality and equality constraints. The sensitivity analysis of the multi-objective function is explicitly carried out by employing the central difference method (CDM), while the solution of the constrained multi-objective optimization problem is sought by making use of the exterior penalty-function method and the golden section method. The numerical experiments are carried out to demonstrate the proposed optimization method and to investigate the optimization results with respect to the porosity distribution pattern and the weighting factors for two single-objective functions. The numerical results inform that the optimum CNT distribution can be successfully sought by the proposed optimization method and the optimization results are remarkably affected by the porosity distribution and the weighting factor.

## 2. Modeling of FG-Porous CNTRC Plate

A CNT-reinforced composite plate is shown in Figure 1a, where length, depth and thickness of the plate are denoted by L, D, and h. Single-walled CNTs (SWCNTs) are aligned along the x-axis and uniformly distributed in the thickness direction. The thickness-wise distribution of CNTs may have a functional gradation as represented in Figure 1b, where four primitive CNT distribution patterns showing different functional gradations are denoted by FG-U, FG-V, FG-O, and FG-X, respectively. In FG-U, the CNT distribution is uniform through the thickness, while the other three have linear variations such that zero at the bottom and the peak at the top in FG-V, zero at the bottom and top and the peak at the mid-surface in FG-O, and zero at the mid-surface and the peak at the bottom and top in FG-X. Being considered as a sort of dual-phase composite material, the effective material properties of FG-CNTRC plates can be predicted using either the modified linear rule of mixture (LRM) or the Mori–Tanaka method [11,21] in which the CNT efficiency parameters $\eta_{j}\left(j=1,2,3\right)$ are introduced. Usually, the modified LRM is widely adopted due to its accuracy and easy application.

The CNTRC plates are usually modeled as an orthotropic material, and their effective elastic and shear modui are estimated by [11]:(1)$$E_{1}=\eta_{1}V_{cnt}E_{1}^{cnt}+V_{m}E^{m},\;\;\frac{\eta_{2}}{E_{2}}=\frac{V_{cnt}}{E_{2}^{cnt}}+\frac{V_{m}}{E^{m}}$$
(2)$$\frac{\eta_{3}}{G_{12}}=\frac{V_{cnt}}{G_{12}^{cnt}}+\frac{V_{m}}{G^{m}}$$
according to the modified LRM. Here, $V_{cnt}$ and $V_{m}=1-V_{cnt}$ indicate the thickness-wise volume fractions of CNTs and polymer. The material properties of the SWCNT and polymer matrix are labeled by the scripts cnt and m, and it is further assumed that $E_{3}=E_{2}$ and $G_{23}^{}=G_{31}^{}=G_{12}^{}$. The scale effect of CNTs on the effective material properties of CNTRCs is taken into consideration by the CNT efficiency parameters $\eta_{j}$ [34], which were determined by matching the effective CNTRC properties obtained by the molecular dynamics (MD) simulation with those predicted by the LRM. Table 1 presents $\eta_{j}$ for the CNTRCs composed of Poly (methyl methacrylate) (PMMA) matrix and CNTs at room temperature.

The volume fraction distributions of CNTs in the above four FG-CNTRCs through the thickness are mathematically expressed by:(3)$$V_{cnt}\left(z\right)=\{\begin{array}{ll} V_{cnt}^{*}, & \;\mathrm{FG}-U\\ \left(1+2z/h\right)V_{cnt}^{*}, & \;\mathrm{FG}-V\\ 2\left(1-2\left|z\right|/h\right)V_{cnt}^{*}, & \;\mathrm{FG}-O\\ 2\left(2\left|z\right|/h\right)V_{cnt}^{*}, & \;\mathrm{FG}-X \end{array}$$
with $V_{cnt}^{*}$ being the total volume fraction of CNTs contained within the polymer matrix. Meanwhile, the effective Poisson’s ratios $\nu_{\alpha \beta }$ and the effective density ρ of CNTRCs are determined in a similar manner:(4)$$\nu_{\alpha \beta }=V_{cnt}^{*}\nu_{12}^{cnt}+V_{m}\nu_{}^{m},\;\alpha \beta =12\to 23\to 31$$
(5)$$\rho =V_{cnt}\rho_{}^{cnt}+V_{m}\rho_{}^{m}$$

Figure 2a represents an FG-porous CNTRC plate in which the porosity density varies in the z-direction only. In the current study, three different porosity distributions are considered: center-biased, lower-biased, and upper-biased, as depicted in Figure 2b. These distributions are called *sym, unsym-1, and unsym-2* in this paper, and those are mathematically repressed by [35]:(6)$$\mathrm{Sym}:\;\psi \left(z\right)=e\cos\left[\pi \left(\frac{z}{h}\right)\right]$$
(7)$$\mathrm{Unsym}-1:\;\psi \left(z\right)=e\cos\left[\frac{\pi }{2}\left(\frac{z}{h}-0.5\right)\right]$$
(8)$$\mathrm{Unsym}-2:\;\psi \left(z\right)=e\cos\left[\frac{\pi }{2}\left(\frac{z}{h}+0.5\right)\right]$$
with e(0≤e≤1) being the porosity parameter. For a given value of e, three porosity distributions have the same porous volume. Owing to the porosity, the equivalent material properties ${℘}_{eff}\left(z\right)$ of the homogenized material model of FG-porous CNTRC plates is modified as:(9)$${℘}_{eff}\left(z\right)={℘}_{eff}\left(z\right)\left(1-\psi \left(z\right)\right)$$
by referring to Phani and Niyogi [36].

Viewing FG-porous CNTRC plates as a 3-D orthotropic body occupying a bounded domain $\Omega \in {ℜ}^{3}$, its displacement field u(x)=u(x,y,z) under the action of body force f and external load $\hat{t}$ is governed by the 3-D elasticity theory:(10)$$\sigma_{ij}{\left(u\right)}_{,j}=f_{i}\;in\;\Omega ,\;ij=x,y,z$$
with the boundary conditions:(11)$$u=\hat{u}\;on\;\Gamma_{D}$$
(12)$$\sigma_{ij}n_{j}={\hat{t}}_{i}\;on\;\Gamma_{N}$$

Here, $\Gamma_{D}$ and $\Gamma_{N}$ stand for the displacement and force boundary regions, and $\sigma_{ij}$ and $n_{j}$ indicate the stress components and the unit normal vector, respectively.

## 3. Analysis and Optimization

### 3.1. Analysis of Bending Deformation

The previous static equilibrium in Equation (10) is converted to the following variational form for solving the bending deformation field:(13)$$\int_{\;V}\varepsilon_{ij}\left(v\right)\sigma_{ij}^{}\left(u\right)dV=\int_{\;S}v^{T}\hat{t}\;dS$$
for every admissible displacement v. Using iso-parametric finite elements, both the trial and test displacement fields u and v are approximated as:(14)$$u=\left[\Phi \right]\left\{\bar{u}\right\},\;v=\left[\Phi \right]\left\{\bar{v}\right\}$$
where [Φ] is a (3×3N) matrix expressed in terms of FE basis functions and $\left\{\bar{u}\right\}$ and $\left\{\bar{v}\right\}$ denote the (3N×1) vectors of nodal displacements.

Next, two matrices [H] and [E] are introduced to compute the strain components $\left\{\varepsilon_{ij}\right\}$ and the stress components $\left\{\sigma_{ij}\right\}$, respectively:(15)$$\left[H\right]=\left[\begin{array}{cccccc} \partial /\partial x & 0 & 0 & \partial /\partial y & 0 & \partial /\partial z\\ 0 & \partial /\partial y & 0 & \partial /\partial x & \partial /\partial z & 0\\ 0 & 0 & \partial /\partial z & 0 & \partial /\partial y & \partial /\partial z \end{array}\right]$$
(16)$$\left[E\right]=\left[\begin{array}{cc} E_{1} & 0\\ 0 & E_{2} \end{array}\right],\;\left[E_{1}\right]=\left[\begin{array}{ccc} C_{11} & C_{12} & C_{13}\\ C_{21} & C_{22} & C_{23}\\ C_{31} & C_{32} & C_{33} \end{array}\right]$$
with $\left[E_{2}\right]=diag\left[G_{12},G_{23},G_{31}\right]$, where $C_{ij}$ indicate the orthotropic material constants [37] expressed in terms of the effective elastic moduli $E_{i}$ in Equation (1) and the effective Poisson’s ratios $\nu_{ij}$ in Equation (4). Then, both the strain and stress components are approximated as:(17)$$\left\{\varepsilon_{ij}\left(v\right)\right\}=\left[B\right]\left\{\bar{v}\right\},\left[B\right]=\left[H\right]\left[\Phi \right]$$
and
(18)$$\left\{\sigma_{ij}^{}\left(u\right)\right\}=\left[E\right]\left[B\right]\left\{\bar{u}\right\}\;$$

Plugging Equations (17) and (18) into the variational form (13), one can derive the simultaneous equation system for solving the plate bending deformation:(19)$$\left[K\right]\left\{\bar{u}\right\}=\left\{F\right\}$$
where the stiffness matrix [K] and the load vector {F} are calculated as:(20)$$\left[K\right]=\int_{\;V}\left[B^{T}\right]\left[E\right]\left[B\right]\;dV$$
(21)$$\left\{F\right\}=\int_{\;S}\left\{\Phi^{T}\right\}\hat{t}\;dS$$

### 3.2. Multi-Objective Optimization of CNT Distribution

For the effective numerical optimization with a reasonable number of design variables, an FG-porous CNTRC beam is divided into (NDV) numbers of uniform homogenized sub-layers, as shown in Figure 3. Then, the CNT volume fractions ${\left(V_{cnt}\right)}_{I}$ of each sub-layer constitutes the design variable vector X such that:(22)$$X=\left\{{\left(V_{cnt}\right)}_{1},\;{\left(V_{cnt}\right)}_{2},\;\ldots ,\;{\left(V_{cnt}\right)}_{NDV}\right\}$$

Owing to the physical constraint, each sub-layer-wise CNT volume fraction ${\left(V_{cnt}\right)}_{I}$ should obey the following lower and upper bounds:(23)$$0\leq {\left(V_{cnt}\right)}_{I}\leq 1,I=1,\;2,\;\ldots ,\;NDV$$
and their arithmetic average should be equal to the preset volume fraction $V_{cnt}^{*}$ of CNTs:(24)$${\left(V_{cnt}\right)}_{1}+{\left(V_{cnt}\right)}_{2}+\cdot \cdot \cdot +{\left(V_{cnt}\right)}_{NDV}=NDV\times V_{cnt}^{*}$$

Letting the vertical displacement $u_{3}$ be w, two single-objective (SO) functions $f_{1}\left(x,X\right)$ are $f_{2}\left(x,X\right)$ are defined by:(25)$$f_{1}\left(X\right)=\underset{x\in \Omega }{\max}\left|\sigma_{eff}\left(x;X\right)\right|,\;f_{2}\left(X\right)=\underset{x\in \Omega }{\max}\left|w\left(x;X\right)\right|$$
with Ω being the entire material domain of the FG-porous CNTRC beam. Then, the multi-objective (MO) function F(X) defined by:(26)$$F\left(X\right)=\max\left(\frac{\left|f_{1}^{}\left(x,X\right)-{\hat{f}}_{1}^{}\right|}{\left|{\hat{f}}_{1}-f_{1}^{*}\right|},\frac{\left|f_{2}^{}\left(x,X\right)-{\hat{f}}_{2}^{}\right|}{\left|{\hat{f}}_{2}-f_{2}^{*}\right|}\right)$$
in terms of the ideal levels $f_{I}^{*}\;\left(I=1,2\right)$ and the aspiration levels ${\hat{f}}_{I}^{}$ of two so functions. The ideal level is the optimum solution obtained by the SO optimization, and the aspiration level indicates the level to be reached by the MO optimization. In the current study, the aspiration level is set by:(27)$${\hat{f}}_{I}=a\cdot f_{I}^{*},\;1.0<a<a^{0},\;I=1,2$$
with $a^{0}=f_{I}^{0}/f_{I}^{*}$, where $f_{I}^{0}$ denote the values obtained by the initial design variable vector $X^{0}$. Note that the relative weights $w_{I}^{f}$ of each SO function is automatically determined such that $w_{I}^{f}=1/\left|{\hat{f}}_{I}-f_{I}^{*}\right|$ once both the ideal and aspiration levels are given. For a detailed explanation, refer to reference [38].

Then, the constrained MO optimization problem of thickness-wise CNT distribution is formulated as follows:(28)$$\mathrm{Find}\;X={\left\{X_{I}\right\}}_{I=1}^{NDV},X_{I}={\left(V_{cnt}\right)}_{I}$$
(29)Minimize F(X)
(30)$$\mathrm{Subject}\mathrm{to}\;\left[K\right]\left\{\bar{u}\right\}=\left\{F\right\}$$
(31)$$h\left(X\right):\sum_{I=1}^{ND}{\left(V_{cnt}\right)}_{I}-NDV\times V_{cnt}^{*}=0$$
(32)$$g_{J}\left(X_{J}\right)\;:\;-{\left(V_{cnt}\right)}_{J}\leq 0,J=1,\;2,\;\ldots ,\;NDV$$
(33)$$g_{J}\left(X_{J-ND}\right)\;:\;{\left(V_{cnt}\right)}_{J-NDV}-1\leq 0,J=NDV+1,\;\ldots ,\;2*NDV$$

It is worth noting that Equation (31) indicates the equality constraint in Equation (24), while Equations (32) and (33) present the inequality constraints in Equation (23), respectively.

The above weighted constrained optimization problem is solved by the exterior penalty-function method (EPFM) [39], which converts the objective function F(X) subject to the constraints to an unconstrained pseudo-objective function $\Phi \left(X;r_{p}\right)$ given by:(34)$$\Phi \left(X;r_{p}\right)=F\left(X\right)+r_{p}c_{1}\left\{h^{2}\left(X\right)\right\}+r_{p}c_{2}\sum_{J=1}^{2*NDV}\max\left[0,g_{J}^{2}\left(X\right)\right]$$
by introducing the exterior penalty parameters $r_{p}$. Here, $c_{1}$ and $c_{2}$ are the normalization factors to keep the balance between the magnitudes of the objective function F(X) and the constraints, which are calculated according to:(35)$$c_{1}=\frac{\max\;\left|\nabla F\left(X\right)\right|}{\max\;\left|\nabla h\left(X\right)\right|},\;c_{2}=\frac{\max\;\left|\nabla F\left(X\right)\right|}{\underset{J}{\max}\;\left|\nabla g_{J}\left(X_{J}\right)\right|}$$

Here, the inequality constraints given in Equations (32) and (33) leads to:(36)$$\left|\nabla g_{J}\left(X_{I}\right)\right|=\left|\partial g_{J}/\partial X_{I}\right|=\left|\pm \delta_{JI}\right|\leq 1$$

So, Equation (33) ends up with:(37)$$c_{3}=\max\left|\nabla F\left(X\right)\right|$$

The optimization iteration starts with a guessed initial design variable $X^{0}$. At each iteration step, the sensitivity analysis is performed, and the convergence is checked using the convergence criterion defined by:(38)$$\left|F\left(X^{k}\right)-F\left(X^{k-1}\right)\right|/\left|F\left(X^{k}\right)\right|\leq \varepsilon_{T}$$

The iteration is terminated when the convergence tolerance $\varepsilon_{T}$ is satisfied; otherwise, it moves to the next iteration by updating the exterior penalty parameter:(39)$$r_{p}^{k+1}=\gamma \cdot r_{p}^{k}$$
where an iteration-independent update constant γ (γ>1) uniformly increases the penalty parameter $r_{p}$ along the optimization iteration. Figure 4 represents the flowchart of the present optimization process.

### 3.3. Sensitivity Analysis

The searching of optimization direction is essential in the mathematical optimization, and it is accomplished by computing the direction vector S of the design variables through the sensitivity analysis. The sensitivity of the pseudo-objective function $\Phi \left(X;c,r_{p}\right)$ with respect to the I−th design variable $X_{I}$ is mathematically expressed by:(40)$$\frac{\partial \Phi \left(X;\;c,\;r_{p}\right)}{\partial X_{I}}=\frac{\partial F\left(X\right)}{\partial X_{I}}+2c_{1}r_{p}h\left(X\right)+2c_{2}r_{p}\left[{\left(V_{cnt}\right)}_{I}-1\right],I=1,\;2,\;\ldots ,NDV$$
according to Equations (25) and (31)–(33). This direct mathematical derivation may be adopted when the thickness-wise CNT distribution is assumed such that the objective function F(X) can be explicitly expressed in terms of design variables ${\left(V_{cnt}\right)}_{I}$. However, even though it can be assumed, the mathematical derivation of the first term on the right-hand side of Equation (40) is highly painstaking.

This direct method can be effectively replaced with the finite difference method (FDM) when the problem size is not so large. In the finite difference method, the direction vector $S^{k}=\left\{S_{1}^{k},\;S_{2}^{k},\;\ldots ,\;S_{ND}^{k}\right\}\;$ at the $k^{th}$ iteration is calculated as:(41)$$S^{k}=\frac{\Phi \left(X^{k-1}+\delta X;\;c_{1}^{k-1},\;c_{2}^{k-1},\;r_{p}^{k-1}\right)-\Phi \left(X^{k-1};\;c_{1}^{k-1},c_{2}^{k-1},r_{p}^{k-1}\right)}{\delta X},k=1,\;2,\;\ldots$$
with the initial exterior penalty parameter $r_{p}^{0}$. Once the direction vector is calculated, the design variable vector is updated according to:(42)$$X^{k}=X^{k-1}+\Delta X^{k},\Delta X^{k}=\beta^{k}S^{k}$$
where $\beta^{k}$ is the iteration-dependent parameter to determine the magnitude of $S^{k}$, and it is calculated by the golden section method [39], which always guarantees the local minimum with respect to the design variable vector X.

## 4. Results and Discussion

Figure 5a shows a simply supported FG-porous CNTRC beam under a uniform vertical distributed load q=0.1 MPa. The beam length L is 0.1 m and the width a and the thickness h are equally 0.01 m, respectively. The porosity parameter e of three cosine-type porosity distributions shown in Figure 2b is set by 0.2. The beam is manufactured with isotropic matrix of Poly (methyl methacrylate) (PMMA) and orthotropic (10,10) single-walled CNTs (SWCNTs). The material properties of constituent materials are given in Table 2, where 1, 2, and 3 indicate x,y and z, respectively. Figure 5b represents ten uniform sub-layers which are divided to suppress the increase of total design variable number by maintaining the flexibility of thickness-wise CNT distribution at the same time. Here, $V_{cnt}^{I}$ indicates the volume fraction of the I-th sub-layer so that the total number of design variables becomes ten. Each sub-layer is uniformly discretized using 100×10 4-node cubic finite elements such that 100 along the x-axis and 10 along the y-axis. So, the whole FG-porous CNTRC beam is uniformly discretized by 10,000 finite elements, and the FE static analysis was carried out by commercial FEM code midas NFX [40].

First, the single-objective optimization for minimizing the peak effective stress is performed with the symmetric porous distribution, for which the initial CNT distribution pattern and the initial CNT volume fractions $V_{cnt}^{I}$ of ten sub-layers are set by FG-U and 0.12. The values of simulation parameters $\varepsilon_{T}$, $r_{p}^{0}$ and γ are set by $1.0\times 10^{-3}$, 1.0 and 2.0, respectively. The beam bending shown in Figure 5a exhibits the edge effect in the stress field near the left and right ends of the beam. Thus, the peak value of stress is extracted from the thickness-wise stress distribution at the mid-span of the beam. The optimization process is terminated in five iterations, as given in Table 3, where the objective function shows a uniform convergence. The peak effective stress occurs simultaneously at the beam top and bottom, and a total of 232 FEM analyses were performed, mostly for the sensitivity analyses. The initial peak effective stress 7.859 MPa is reduced to 5.985 MPa at the final stage, and the reduction amount is 1.874 MPa, which corresponds to 23.8% of the initial peak effective stress.

Figure 6a comparatively represents the initial and optimum CNT distributions, where the optimum CNT distribution for a non-porous (i.e., dense) CNTRC beam is added for comparison purposes. The arithmetic average of layer-wise CNT volume fractions $V_{cnt}^{I}$ is found to be 0.1208 so that the equality constraint in Equation (31) is strictly satisfied. One can see the difference in CNT distributions between porous and non-porous cases, and the difference becomes more apparent at the central region. This is because the porosity in the symmetric porosity distribution is dominated in the central region, as depicted in Figure 2b. Figure 6b comparatively represents the thickness-wise effective stress distributions of initial and optimum CNT distributions. The initial one is symmetric and shows a linear variation from zero at the mid-surface to the peak value at the beam bottom and top. In the bending of a homogeneous beam, the linearly varying bending strain produces the linear bending stress distribution through the thickness. Meanwhile, the optimal CNT distribution also leads to a symmetric effective stress distribution with zero at the mid-surface, but its distribution is not linear anymore and its peak is smaller than that of the initial CNT distribution. This is because the parabolic-type distribution of CNTs and porosity leads to the parabolic distribution of elastic modulus, which leads to the non-linear stress distribution with smaller peak effective stress. The difference in the effective stress distribution between porous and non-porous cases is not so remarkable.

The numerical optimization was also performed for two different porosity distributions, *unsym-1* and *unsym-2*, and the optimization results are summarized in Table 4. The total numbers of iteration and FEM analyses of *unsym-1* and *2* are smaller than those of sym, but *unsym-1* and *2* show the same total numbers in iteration and FEM analyses. The peak effective stress occurs at the top of the beam for *unsym-1* while at the bottom for *unsym-2*, but its magnitude is shown to be the same for both cases. When compared with the case of sym, *unsym-1* and *2* lead to smaller initial and final effective stresses. Thus, it is found that there exists a remarkable difference in the CNT distribution between symmetric and unsymmetric porous distributions, but two unsymmetric porous distributions show the difference only in the location of peak effective stress.

Figure 7a,b comparatively represent the optimum CNT distributions between *unsym-1* and *unsym-2* porosity distributions. It is observed that both distributions of *unsym-1* and *2* are exactly anti-symmetric with respect to the mid-surface. This anti-symmetry is solely owing to the anti-symmetry in their porosity distributions. When compared with the optimum CNT distribution of sym, *unsym-1* shows a slightly larger difference in the lower beam region while *unsym-2* leads a slightly larger difference in the upper beam region. This is because the porosity in *unsym-1* is dominated in the lower region but that in *unsym-2* is dominated in the upper region, as depicted in Figure 2b.

Figure 8a compares the effective stress distributions of three porosity distributions for the optimal CNT distribution. The zero line in the effective stress distribution of *unsym-1* moves slightly upwards, and vice versa for *unsym-2*. This is because the porosity dominance in the lower beam region for *unsym-1* gives rise to higher elastic modulus in the upper beam region, and vice versa for *unsym-2*. However, it is seen that the relative difference in the magnitudes of effective stress is not remarkable. Figure 8b represents the dependence of iteration history of objective function on the porosity distribution. *unsym-1* and *2* show exactly the same iteration history because both porosity distributions produce exactly anti-symmetric stress distributions through the thickness. Except for the relative difference in the effective stress magnitude, both symmetric and unsymmetric porosity distributions show rapid and stable convergence.

Next, the multi-objective optimization for minimizing the peak effective stress and the peak deflection was performed for the sym porosity distribution using the same FG-porous CNTRC beam shown in Figure 5a. The aspiration levels in Equation (27) for two SO functions $f_{1}$ and $f_{2}$ were set by $s_{1}=s_{2}=1.2$, while the other simulation parameters were kept the same with the previous single-objective optimization. For the sym porosity distribution, the ideal levels $f_{1}^{*}$ and $f_{2}^{*}$ were 5.98457 MPa and 0.03174 *mm* which were obtained by SO optimization for the peak effective stress and for the peak deflection, respectively. Thus, the values taken for two aspiration levels become ${\hat{f}}_{1}^{}=s_{1}f_{1}^{*}=7.18148\;MPa$ and ${\hat{f}}_{2}^{}=s_{2}f_{2}^{*}=0.03809\;mm$, respectively. As will be presented later in detail, the SO optimization for minimizing the peak deflection was terminated in five iterations, and the initial peak deflection of 0.05229 mm was reduced to 0.03174 mm.

Table 5 presents the variation of MO function F(X) to the iteration, where the objective function decreases with small fluctuation and the optimization terminates in six iterations with a total of 271 FEM analyses. The peak effective stress occurs initially at the beam top and bottom, but the peak location moves to the inside of the beam during the next three iterations, and thereafter the peak effective stress occurs again at the beam top and bottom. The optimal CNT distribution is represented in Figure 9a, where those obtained by two SO optimizations are also given for comparison purposes. Here, ${\mathrm{SO}}_{-\sigma }$ and ${\mathrm{SO}}_{-w}$ denote the SO optimizations for the peak effective stress and for the peak deflection, respectively. It is clearly observed that ${\mathrm{SO}}_{-\sigma }$ and ${\mathrm{SO}}_{-w}$ lead to the optimal CNT distributions showing the opposite thickness-wise variations. The former leads to smaller CNT volume fraction at the beam top and bottom to decrease the peak effective stress by decreasing the elastic modulus, while the latter leads to larger CNT volume fraction at the beam top and bottom to decrease the peak deflection by increasing the beam bending stiffness. Meanwhile, MO leads to the optimal CNT distribution which is placed between those of ${\mathrm{SO}}_{-\sigma }$ and ${\mathrm{SO}}_{-w}$. This result was obtained because MO tries to minimize the peak effective stress and the peak deflection at the same time. This trend of MO is also observed from Figure 9b, where the thickness-wise effective stress distribution of MO is placed between those of ${\mathrm{SO}}_{-\sigma }$ and ${\mathrm{SO}}_{-w}$. Note that the order in the magnitude of peak effective stress is ${\mathrm{SO}}_{-w}$ > MO > ${\mathrm{SO}}_{-\sigma }$, which is consistent with the minimization goals of ${\mathrm{SO}}_{-w}$, MO, and ${\mathrm{SO}}_{-\sigma }$.

The iteration histories of $\sigma_{eff}^{\max}$ and $w_{}^{\max}$ between MO and SO optimizations are compared in Table 6 and represented in Figure 10. First of all, it is found that $\sigma_{eff}^{\max}$ and $w_{}^{\max}$ obtained by the MO optimization are larger than those obtained by the SO optimizations. This is consistent with the fact that the optimum solution of MO optimization is always larger than the optimum solution of SO optimization. The total number of iterations of MO is longer than one of SO because the MO optimization exhibits larger fluctuation than the SO optimization, as can be realized from the comparison of Table 3 and Table 5.

Next, the multi-objective optimization was performed by changing the aspiration levels $s_{1}$ and $s_{2}$ for two single-objective functions $f_{1}$ and $f_{2}$. The optimization results of two combinations of aspiration levels, which are $\left(s_{1}:s_{2}\right)=$ (1.3:1.1) and (1.1:1.3), are presented in Table 7, where the results of the previous case (1.2:1.2) are also given for the purpose of comparison. The case of (1.3:1.1) shows the total numbers of iteration and FEM analyses which are similar to those of the case of (1.2:1.2), but the case of (1.1:1.3) leads to smaller total numbers in the optimization iteration and FEM analyses. Meanwhile, it is found that the case of (1.3:1.1) leads to the optimum solution equal to F(X)=0.66486, which is smaller than those of (1.2:1.2) and (1.1:1.3). This is because the case of (1.3:1.1) places more weight on the reduction of peak deflection, and the optimum value of F(X) was determined from the relative reduction in the peak deflection, as given in Equation (26). This can be confirmed from the fact that the case of (1.3:1.1) leads to smaller peak deflection $w^{\max}=0.03248\;mm$ but larger peak effective stress $\sigma_{eff}^{\max}=7.17380\;MPa$ compared with the previous case of (1.2:1.2). On the other hand, the case of (1.1:1.3) places more weight on the reduction in the peak effective stress, so that it leads to smaller peak effective stress $\sigma_{eff}^{\max}=6.08067\;MPa$ but larger peak deflection $w^{\max}=0.04219mm$ compared with the case of (1.2:1.2). Thus, it has been justified that the trade-off between the peak effective stress and the peak deflection is successfully accomplished by adjusting the aspiration levels (i.e., the combination of ($s_{1}:s_{2}$)).

Figure 11a comparatively represents the optimal CNT distributions for three different combinations of aspiration levels. When compared with the case of (1.2:1.2), the case of (1.1:1.3) leads to the distribution in which the CNT volume fraction $V_{cnt}^{}$ is high in the central region while low in the vicinity of top and bottom. This is because the peak effective stress occurring at the beam top and bottom can be effectively reduced by reducing the elastic modulus of the beam in the vicinity of the beam top and bottom, and the reduction of elastic modulus can be made by decreasing the CNT volume fraction. Meanwhile, the case of (1.3:1.1) leads to the optimum CNT distribution which is almost opposite to that of the case of (1.1:1.3). This is because the beam deflection can be reduced by increasing the bending flexural rigidity, and the flexural rigidity can be effectively increased by placing more CNTs in the vicinity of the beam top and bottom. Figure 11b compares the thickness-wise effective stress distributions, where the case of (1.3:1.1) shows the highest level while the case of (1.1:1.3) leads to the lowest level. Meanwhile, the effective stress distribution of the case of (1.2:1.2) passes through between those of the cases of (1.1:1.3) and (1.3:1.1). This comparison justified again the trade-off between $\sigma_{eff}^{\max}$ and $w^{\max}$ according to the adjustment of aspiration levels.

## 5. Conclusions

The multi-objective optimization of FG-CNTRC porous beams was presented to simultaneously minimize the peak effective stress and the peak deflection. An FG-CNTRC porous beam was divided into uniform sub-layers, and the layer-wise uniform CNT volume fractions were chosen as the design variables. The peak effective stress and the peak deflection were normalized using their ideal and aspiration levels which were obtained by the single-objective optimization and set by the designer’s decision making. The maximum value between two normalized quantities was defined as the multi-objective function, and it was minimized by making use of the exterior penalty-function method and the trade-off process. The proposed method was demonstrated and verified through the benchmark experiment, and the optimization solutions were investigated with respect to the porosity distribution pattern and the combination of aspiration levels. The numerical results provide the following major observations:The proposed MO optimization method successfully seeks the optimum CNT distribution, which simultaneously reduces the peak effective stress and the peak deflection with the stable convergence.The optimal CNT distribution of a porous beam is different from that of a non-porous beam, and the difference is apparent in the region where the porosity is dominant. However, the effect of porosity on the optimum effective stress distribution is not so remarkable.The porosity distribution pattern significantly affects the optimum CNT distribution but its effect on the optimum effective stress distribution is insignificant. *unsym-1* and *2* porosity distributions lead to the optimum CNT distributions, which are anti-symmetric to each other.The MO optimization leads to the optimal CNT distribution and the optimal effective stress distribution, which are placed between those of ${\mathrm{SO}}_{-\sigma }$ and ${\mathrm{SO}}_{-w}$. The order in the magnitude of $\sigma_{eff}^{\max}$ is ${\mathrm{SO}}_{-w}$ > MO > ${\mathrm{SO}}_{-\sigma }$, and that of $w_{}^{\max}$ is ${\mathrm{SO}}_{-\sigma }$> MO > ${\mathrm{SO}}_{-w}$.The trade-off between the peak effective stress and the peak deflection can be successfully accomplished according to the adjustment of aspiration levels such that the decrease of $s_{I}$ places more weight on the corresponding SO function $f_{I}$.Regarding the trade-off between two SO functions, the case of (1.1:1.3) leads to the lowest peak effective stress while the case of (1.3:1.1) provides the smallest peak deflection.


## Figures and Tables

**Figure 1 polymers-15-00349-f001:**
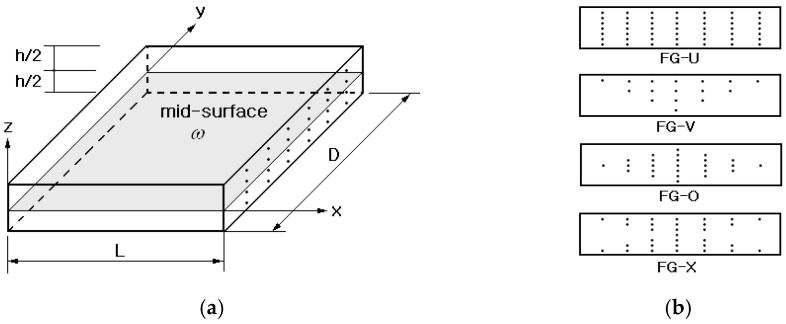
A CNT-reinforced composite (CNTRC) plate: (**a**) geometry and dimensions, (**b**) four functionally graded (FG) CNT distributions.

**Figure 2 polymers-15-00349-f002:**
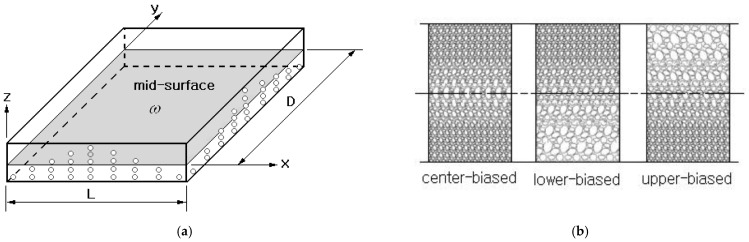
FG-porous CNTRC plate: (**a**) thickness-wise porosity gradient, (**b**) three different porosity distributions.

**Figure 3 polymers-15-00349-f003:**
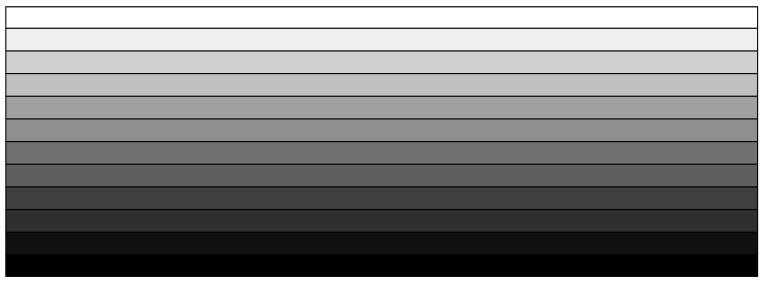
Uniform division of an FG-porous CNTRC beam into homogenized sub-layers.

**Figure 4 polymers-15-00349-f004:**
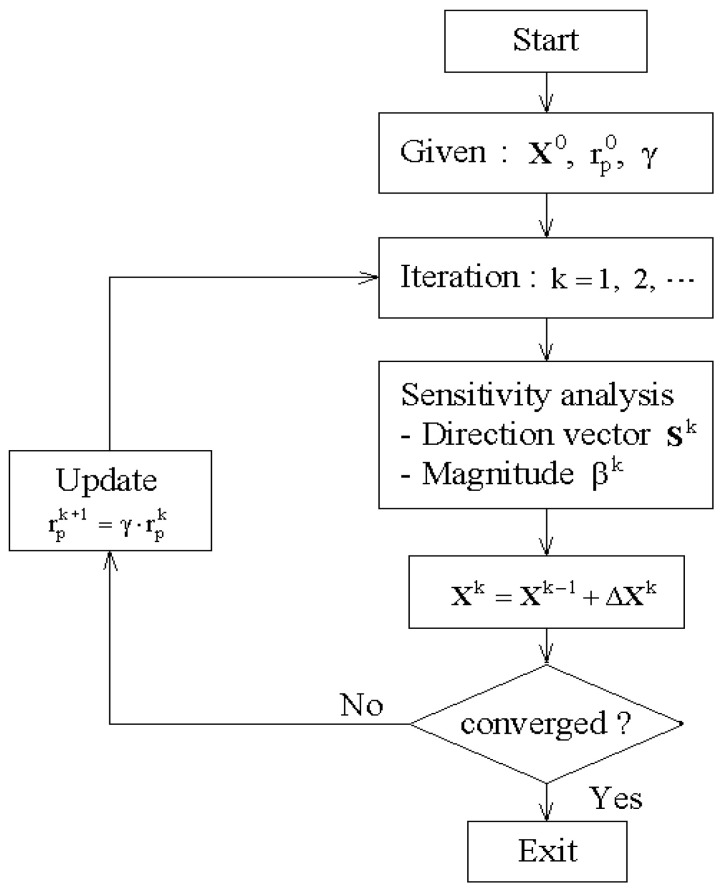
A flowchart of the CNT distribution optimization.

**Figure 5 polymers-15-00349-f005:**
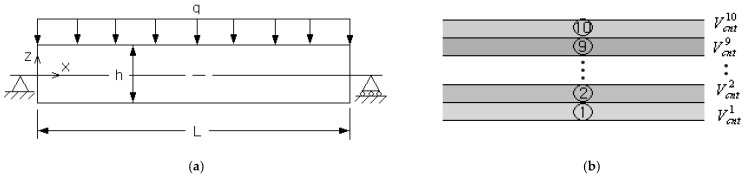
A simply supported FG-porous CNTRC beam: (**a**) dimensions and loading/boundary conditions, (**b**) uniform sub-layers.

**Figure 6 polymers-15-00349-f006:**
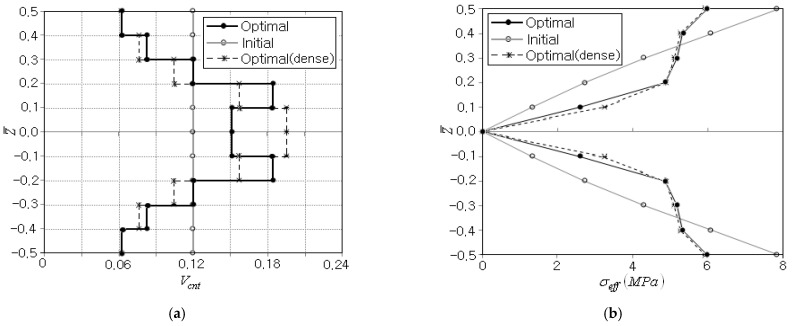
Comparison of initial and final distributions through the thickness: (**a**) the CNT volume fraction $V_{cnt}^{*}$, (**b**) the effective stress $\sigma_{eff}$.

**Figure 7 polymers-15-00349-f007:**
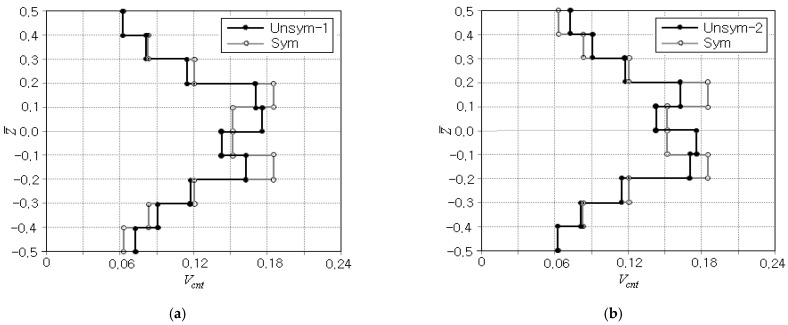
Optimal distributions of CNT volume fraction $V_{cnt}^{*}$: (**a**) *unsym-1*, (**b**) *unsym-2*.

**Figure 8 polymers-15-00349-f008:**
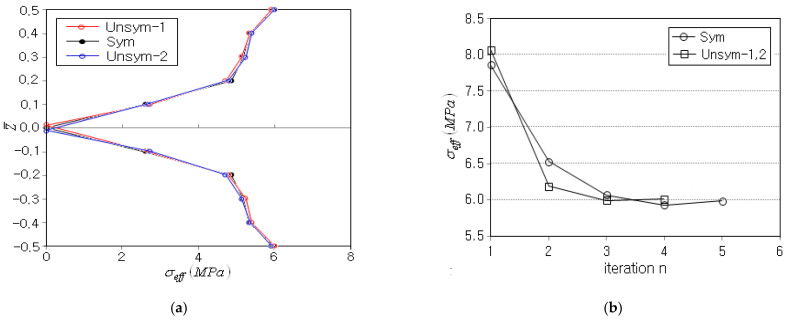
Comparison: (**a**) effective stress distributions $\sigma_{eff}$ for the optimal CNT distribution, (**b**) iteration histories of the objective function.

**Figure 9 polymers-15-00349-f009:**
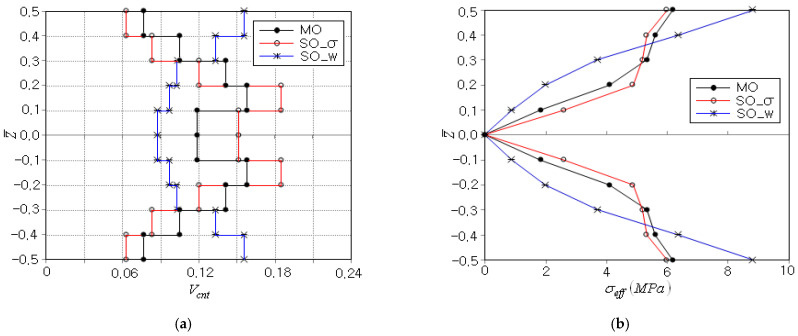
Comparison between MO and SO optimizations: (**a**) optimal CNT distributions, (**b**) thickness-wise effective stress distributions.

**Figure 10 polymers-15-00349-f010:**
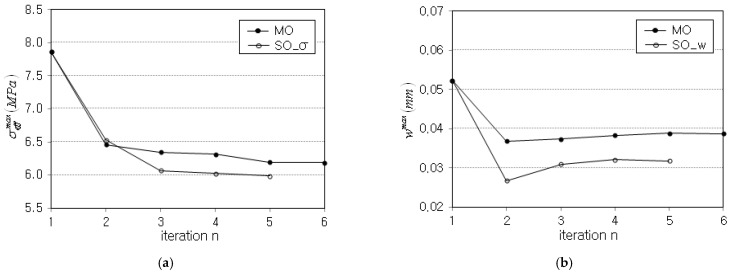
Comparison of iteration histories: (**a**) peak effective stress $\sigma_{eff}^{\max}$, (**b**) peak deflection $w_{}^{\max}$.

**Figure 11 polymers-15-00349-f011:**
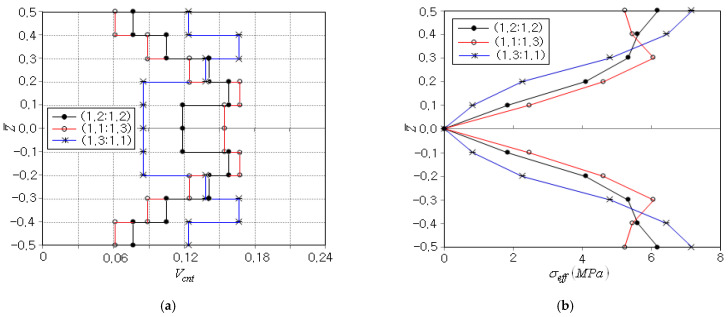
Trade-off between $\sigma_{eff}^{\max}$ and $w^{\max}$: (**a**) optimal CNT distribution, (**b**) thickness-wise effective stress distribution.

**Table 1 polymers-15-00349-t001:** The CNT efficiency parameters $\eta_{j}$ for three different values of $V_{cnt}^{*}$ (PMMA/CNT at T=300 K).

$V_{cnt}^{*}$	$\eta_{1}$	$\eta_{2}$	$\eta_{3}$
0.12	0.137	1.022	0.715
0.17	0.142	1.626	1.138
0.28	0.141	1.585	1.109

**Table 2 polymers-15-00349-t002:** Material properties of SWCNT and matrix (PMMA) [41].

Materials	Young’s Modulus (GPa)			Poisson’s Ratio			Shear Modulus (*GPa*)			Density $(kg/m^{3}$)
	$E_{1}$	$E_{2}$	$E_{3}$	$\nu_{12}$	$\nu_{23}$	$\nu_{31}$	$G_{12}$	$G_{23}$	$G_{31}$	ρ
SWCNT	5646.6	7080.0	-	0.175	-	-	1944.5	-	-	1400
PMMA	2.5			0.34			0.9328			1150

**Table 3 polymers-15-00349-t003:** Variation of the objective function to the iteration (sym).

Iteration	$\mathrm{Objective}\;\mathrm{Function}\;\sigma_{eff}^{\max}$	Location (z)
Initial	7.85894 $\times 10^{6}\;Pa$	±5.0
2	6.53026 $\times 10^{6}\;Pa$	±5.0
3	6.06824 $\times 10^{6}\;Pa$	±5.0
4	5.92605 $\times 10^{6}\;Pa$	±5.0
5	5.98457 $\times 10^{6}\;Pa$	±5.0
Total number of FEM analyses		232

**Table 4 polymers-15-00349-t004:** The optimization results for three different porosity distributions (e=0.2).

Items	$$\mathrm{CNT}\;\mathrm{Volume}\;\mathrm{Fractions}\;V_{cnt}^{*}$$		
	Sym	*unsym-1*	*unsym-2*
$$\mathrm{Initial},\;\sigma_{eff}^{\max}\left(X_{0}\right)\;\left(MPa\right)$$	7.85894	8.06819	8.06819
$$\mathrm{Optimum},\;\sigma_{eff}^{\max}\left(X_{opt}\right)\;\left(MPa\right)$$	5.98457	6.01258	6.01258
$$\mathrm{Location}\;\left(\bar{z}\right)$$	±0.5	+0.5	−0.5
Iterations	5	4	4
FEM analyses	232	151	151

**Table 5 polymers-15-00349-t005:** Variation of MO function to the iteration (sym, $s_{1}=s_{2}=1.2$ ).

Iteration	Multi-Objective Function F(X)	Location (z $)\;\mathrm{of}\;\sigma_{eff}^{\max}$
Initial	3.23855	±5.0
2	0.80688	±3.0
3	0.88217	±3.0
4	1.03030	±4.0
5	1.10915	±5.0
6	1.09947	±5.0
Total number of FEM analyses		271

**Table 6 polymers-15-00349-t006:** Comparison of iteration histories between MO and SO optimizations.

Methods	Objective Functions	Iteration					
		1	2	3	4	5	6
MO optimization	$\sigma_{eff}^{\max}\left(MPa\right)$	7.85894	6.45611	6.34687	6.31065	6.19077	6.18648
	$w_{}^{\max}\left(mm\right)$	0.05229	0.03686	0.03734	0.03828	0.03878	0.03872
SO optimization	$\sigma_{eff}^{\max}\left(MPa\right)$	7.85894	6.53026	6.06824	5.92605	5.98457	-
	$w_{}^{\max}\left(mm\right)$	0.05229	0.02675	0.03097	0.03204	0.03174	-

**Table 7 polymers-15-00349-t007:** The optimization results for three different combinations of aspiration levels.

Items	$\mathrm{Combinations}\;\mathrm{of}\;\mathrm{Aspiration}\;\mathrm{Levels}\;(s_{1}:s_{2})$		
	(1.2:1.2)	(1.3:1.1)	(1.1:1.3)
F(X)	1.09947	0.66486	1.09756
$f_{1}\left(X\right)=\sigma_{eff}^{\max}\left(MPa\right)$	6.18648	7.17389	6.08067
$f_{2}\left(X\right)=w_{}^{\max}\left(mm\right)$	0.03872	0.03248	0.04219
Iterations	6	6	5
FEM analyses	271	270	193

## Data Availability

Not applicable.

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
