# Peer review of "Optimal Tailoring of CNT Distribution in Functionally Graded Porous CNTRC Beams"

_polymers, 2023, doi:10.3390/polym15020349_

Round 1

Reviewer 1 Report

There are recent studies on FGM in the literature. Please support your results with literature studies or guide the reader by converting them to dimensionless (Free vibration analysis of carbon nanotube RC nanobeams with variational approaches; A refined functional and mixed formulation to static analyses of fgm beams; Optimization of flexure stiffness of FGM beams via artificial neural networks by mixed FEM; Free vibration analysis of open-cell FG porous beams: analytical, numerical and ANN approaches).

Where did you get the material properties?

According to which theory did you determine the kinematic displacement relations?

Author Response

Please refer to the response to reviewers' comments attached.

Reviewer 2 Report

The manuscript presents a multi-objective optimization method for applications in the material design with minimizing stress and mechanical deflection in polymer composites. The application to CNT (carbon nanotubes) reinforced polymer composites is demonstrated through numerical solutions. In order to adequately represent in their model the actual composite material property, the authors have incorporated in their approach not only the different CNT distributions within the polymer matrix, but also it introduced the material parameter called “porosity”. The manuscript presents the step-by-step numerical evaluation of the optimization process for a composite beam made of CNT and PMMA with different porosity distributions. The role of the porosity profile and the CNT distributions on mechanical beam characteristics was shown with sufficient details. I would recommend the manuscript for publishing with minor corrections.  

Comments

The abbreviations FG-U, FG-V, FG-O and FG-X on page 2, need explanations such referencing to Fig.1(b).

It has remained unclear. What CNT distribution pattern (V_cnt(z)) have you assume in the numerical analysis (section 4)? It seems the CNT distribution profile is a variable parameter. Please state it clearly.

Eq (9), “the equivalent material properties” is unclearly described. Please specify in connection with your approach.

Author Response

(The authors gave the same response as above.)

Reviewer 3 Report

The abstract should briefly cover the main conclusions, and potential significance/applications of this research work. Currently, it fails to do this.

The language needs to be polished, to name a few: ‘…the extensive research efforts have focused to investigate…’, should be ‘…focused on investigating…’

Include a bit more introduction of porous FGMs, such as doi: 10.1016/j.compositesb.2015.07.018; 10.1080/10426914.2022.2075892

Author Response

(The authors gave the same response as above.)

Round 2

Reviewer 1 Report

The authors made the necessary revisions. The article is acceptable as it is.